# CNS Ageing in Health and Neurodegenerative Disorders

**DOI:** 10.3390/jcm12062255

**Published:** 2023-03-14

**Authors:** Evangelia Kesidou, Paschalis Theotokis, Olympia Damianidou, Marina Boziki, Natalia Konstantinidou, Charilaos Taloumtzis, Styliani-Aggeliki Sintila, Panagiotis Grigoriadis, Maria Eleptheria Evangelopoulos, Christos Bakirtzis, Constantina Simeonidou

**Affiliations:** 1Laboratory of Experimental Neurology and Neuroimmunology, 2nd Department of Neurology, AHEPA University Hospital, Aristotle University of Thessaloniki, 546 36 Thessaloniki, Greeceptheotokis@auth.gr (P.T.);; 2Laboratory of Physiology, Faculty of Medicine, Aristotle University of Thessaloniki, 541 24 Thessaloniki, Greece; 3First Department of Neurology, Aeginition Hospital, National and Kapodistrian University of Athens, 115 28 Athens, Greece

**Keywords:** ageing, immunosenescence, inflammageing, CNS, plasticity, DMTs

## Abstract

The process of ageing is characteristic of multicellular organisms associated with late stages of the lifecycle and is manifested through a plethora of phenotypes. Its underlying mechanisms are correlated with age-dependent diseases, especially neurodegenerative diseases such as Alzheimer’s disease (AD), Parkinson’s disease (PD) and multiple sclerosis (MS) that are accompanied by social and financial difficulties for patients. Over time, people not only become more prone to neurodegeneration but they also lose the ability to trigger pivotal restorative mechanisms. In this review, we attempt to present the already known molecular and cellular hallmarks that characterize ageing in association with their impact on the central nervous system (CNS)’s structure and function intensifying possible preexisting pathogenetic conditions. A thorough and elucidative study of the underlying mechanisms of ageing will be able to contribute further to the development of new therapeutic interventions to effectively treat age-dependent manifestations of neurodegenerative diseases.

## 1. Introduction

One of the most fascinating enigmas that intrigues people throughout their existence is the progressive biological phenomenon which sets limits upon the lifespans of living organisms and is known as ageing. Although ageing has instigated an interest to be decoded by the scientific community, it is still not easily defined. Ageing is characterized as the time-related functional deterioration that normally takes place after sexual maturity. It has been accused as the main cause of the onset of various chronic diseases; thus, a better understanding and exploration of the underlying mechanisms may allow for targeted interventions and manipulation with exogenous factors in favor of health restoration [1,2].

Ageing is tightly associated with a decline in cognitive and motor functions; nevertheless, people exhibit a wide range of phenotypes as some of them develop diseases with time progression, while others remain healthy and highly active even at 100 years of age [3,4]. The most common manifestations of ageing are low muscle strength and control, which are linked to high cardiovascular disease risk and cognitive impairment, a multifactorial process that refers to memory, conceptual reasoning and vocabulary processing speed [5], and can be manageable if detected early [1]. Other changes that occur during ageing are a loss of bone tissue, the replacement of muscle with fat, irregularities in hormone levels and vascular changes, which can affect mainly the heart and the brain [6]. In particular, changes in neurons and synapses may be associated with mental, metabolic and circadian rhythm modifications [5,7] and could possibly evolve into fatal neurodegenerative disorders [8].

A recent multi-omic analysis proceeded with the identification of ageing patterns in individuals of different ages, termed as ageotypes, which were in accordance with the molecular profiling that the individuals exhibited over time. The same study also revealed four predominant ageotypes correlated with kidney, liver, metabolic and immune pathways, which are involved in various chronic diseases as well [9,10]. Therefore, considering that ageing is affected by the habits (e.g., tobacco and alcohol consumption, lifestyle choices) and health condition of a person, ageotypes may also be applicable in personal ageing evaluation, monitoring and intervention [9]. 

The etiology of ageing remained a mystery for a long time and numerous theories were proposed to delineate the details of the whole process. Conventional theories supported the notion that ageing is not related to adaptation or genetic programming [11]. On the other hand, according to recent programmed and damage-based theories, ageing is characterized by a sequence of biological events modulated by genetic and environmental factors, respectively, that lead to cumulative impairments [1,11]. Multiple factors seem to be involved synergistically in the initial stages of this phenomenon rather than a single one, such as reactive oxygen species (ROS) or free radicals [12]; thus, regulating a few of them could remarkably extend one’s lifespan [13]. 

Several hallmarks have been described in an attempt to identify the underlying molecular and cellular mechanisms of ageing that eventually result in cognitive and physical impairment. Specifically, age-related diseases have been associated with genetic damages caused by environmental or intrinsic factors, such as ROS, as well as with epigenetic changes and telomere shrinkage [2]. Moreover, a loss of protein homeostasis has been identified as a characteristic of ageing and neurodegenerative diseases such as Alzheimer’s disease (AD), in addition to the mutations that are linked to dysregulated nutrient sensing and other metabolic pathways [2,14]. Over time, age-related factors seem to lead cells to surrender to senescence [15], acquiring a pro-inflammatory secretome that fuels ageing mechanisms to a greater extent [16]. Gradually, organelles such as mitochondria lose their normal and regenerative functions [2,17], diminished immune cells lead to immunosenescence and the whole organism is subjected to inflammation while immunosurveillance is compromised [2]. A better understanding of ageing pathogenetic mechanisms, therefore, is essential in order to interfere with the responsible pathways and effectively treat age-dependent neurological manifestations [7].

## 2. Ageing at the Molecular and the Cellular Level

Ageing hallmarks and underlying mechanisms can be subdivided into a handful of categories, led by the cause of cellular damage, the response to it and the phenotypic activity based on the molecular abnormalities. 

### 2.1. Ageing Damaging Mechanisms

DNA instability, epigenetic alterations, telomere shrinkage and proteostasis (protein homeostasis) disturbance can collectively comprise the primary hallmarks [18]. Mutagens, such as chemicals and radiation, can cause large chromosomal rearrangements or point mutations, which accumulate with time rendering cells more prone to genetic instability which is linked to age-related disorders [18]. Notably, genomic instability has been associated with the normal as well as the pathogenic ageing of the human brain and it may be a factor for the development of neurodegenerative diseases such as AD [19] and PD [20]. Epigenetic alterations, which are acquired modifications in gene expression including DNA histone modifications and gene silencing via non-coding RNAs, such as microRNAs [21] and methylation, mainly in CpG islands that had been associated with a shorter lifespan [22] appear more frequently in the elderly [23,24]. Moreover, telomeres, repetitive nucleotide sequences at the end of chromosomes, become unstable and shrink over time resulting in the manifestation of premature ageing syndromes as well as a decreased lifespan [25]. 

The maintenance of protein homeostasis which is governed by molecular chaperones, the proteasome and the process of autophagy can be dysregulated, leading to senescence. In later life, the accumulation of damaged proteins and molecules such as lipofuscin along with impaired DNA repair systems can affect cell survival and lead to age-dependent diseases such as AD [8,26,27,28] and PD [29,30]. Notably, autophagy, which takes part in recycling damaged cell components, seems to decline with age and its upregulation can reduce the possibility of diseases and a shorter lifespan [21]. Autophagy is a pivotal cellular mechanism; thus, its dysregulation can affect, amongst others, the immune and nervous systems leading to correlated diseases [21,31]. 

### 2.2. Responses to Damage

The secondary ageing hallmarks include mitochondrial dysfunction and deregulation in nutrient sensing, which both can lead to a gradual accumulation of senescent cells that escape the declined immune surveillance, increasing vulnerability to infections [2,21,32,33]. Mitochondria are highly affected by constant exposure to ROS and since ROS can oxidize their proteins and other macromolecules, mitochondrial dysfunction has been linked to various age-dependent diseases [34]. Analyses of human, rat and rhesus macaque brains have shown a reduced expression of mitochondrial genes over time, while their function can decrease or increase one’s lifespan under certain circumstances [35]. As a matter of fact, mutations in mtDNA can lead to human mitochondrial encephalomyopathy development, which is characterized by neurological and muscle impairments depending on the number of damaged mitochondria [35]. An updated insight into mitohormesis, the response of mitochondria to external stimuli, shows that ROS can be tolerable up to a certain threshold level and act as signaling cues along with adenosine monophosphate (AMP) and nicotinamide adenine dinucleotide (NAD), regulating cellular homeostasis and extending longevity [36]. More specifically, low AMP and NAD levels correlate with metabolic and energy deprivation, especially in cells with high energy consumption, such as neuronal and muscle cells, thus leading to senescent-associated neurodegenerative diseases and muscle atrophy [37]. 

It has been long accepted that AMP-activated protein kinase (AMPK) is a major component of mitochondrial recycling and de novo synthesis following energy deprivation [38]. The two central mediators of nutrient sensing are the mammalian target of rapamycin TOR (mTOR) and its inhibitor, AMPK, that is activated via exercise and calorie restriction. Evidence from calorie restriction in animals, looking into insulin insensitivity and the highly conserved insulin and IGF-1 pathway animal models, support the notion of a prolonged lifespan in these animals [39,40]. Increased AMP levels are also linked to high levels of NAD that activate the so-called longevity genes, such as sirtuins (SIRTs) [21]. Sirtuins are highly conserved NAD-dependent deacetylases that regulate ageing and longevity [41], interacting with mTOR and the insulin–forkhead box protein O (FOXO) as well [5]. Among its functions, SIRT1 can lead to alterations in histone patterns and gene silencing in response to DNA damages and ageing [35]. Other than that, SIRT1 can promote dendrite and axon growth, regulate learning, memory and the circadian rhythm and act synergetically with SIRT2-regulating neural stem cells [5]. 

### 2.3. Alterations That Affect Phenotypic Manifestations 

The last ageing hallmarks, which reflect the primary and secondary aforementioned traits, refer to apoptosis, stem cell depletion and unbalanced intercellular communication. Apart from being involved in the elimination of neurons in early life, apoptosis also occurs during neuronal death that takes place in normal ageing and is enhanced by increased pro-apoptotic proteins such as PARP and caspase-3 in the brain [8]. Additionally, high ROS levels and Ca2+ influx through receptors such as AMPA (α-amino-3-hydroxy-5-methyl-4-isoxazole propionic acid) and NMDA (*N*-methyl-D-aspartate) can lead to dendrites’ destruction or cell death mediated by excitotoxicity in older people [8]. 

A loss of stem cell functions usually occurs in ageing leading to its characteristic phenotypes such as sarcopenia where muscle stem cells are dysregulated [21], anemia and osteoporosis [2]. Another significant example is the limited function and regeneration of oligodendrocyte progenitor cells (OPCs) in the CNS due to OPCs’ niche stiffness during ageing [42]. Notably, OPCs’ inability to produce oligodendrocytes, due to DNA damages or low metabolic function, has been correlated with severe neurodegeneration in multiple sclerosis (MS); thus, OPCs’ regeneration has been a target of treatments for chronic demyelination [43]. The inhibition of the mechanoresponsive ion channel Piezo1-mediated signaling appears to restore OPCs’ proliferation and differentiation ability in aged rats [42]. Fasting or metformin administration also seem to favor the restoration of OPCs’ regeneration and differentiation capacity, promoting remyelination in aged animals [43] where it may be disrupted due to inefficient epigenetic regulation of gene expression [44]. The latter might occur because of inefficacious recruitment of histone deacetylases (HDACs), leading to transcriptional inhibitors gathering and interference with myelin gene expression that results in a loss of OPCs’ ability to proliferate [44]. Remyelination can also be restricted due to a failure to cope with inflammatory responses to myelin damage and an inability to remove cholesterol properly [45]. Thus, myelin debris increases, forming cholesterol crystals in aged phagocytes whose membrane destruction activates inflammasomes restraining regeneration in the tissue [45]. 

Of note, cholesterol, that is produced in the CNS mainly by oligodendrocytes and by astrocytes, constitutes up to 80% of the myelin membranes while its availability regulates remyelination. During ageing, apart from the dysregulation of oligodendrocyte, astroglial and microglial cell function, major causes of decreased remyelination, reduced cholesterol synthesis occurs. A lower expression of cholesterol-producing enzymes due to aged astrocytes and OPCs results in a decrease in cholesterol synthesis rates and cholesterol content in the ageing myelin membranes. Ineffective liver X receptor (LXR) signaling further disturbs lipid recycling and hinders remyelination [46]. 

Over time, various alterations happen in neuronal and endocrine communication and in the immune system which are termed as immunosenescence. The most common consequences regarding age and the less robust immune system are the decreased levels of antibody production and thymus atrophy, the higher levels of differentiated and senescent T cells, which do not proliferate and have a senescence-associated secretory phenotype (SASP) rich in pro-inflammatory cytokines [47,48], and less naive cells to confront the wide variety of pathogenic invaders [49,50]. The SASP factors are linked to senescence induction with p14ARF/p19ARF, p16INK4 and p21 being established as important ageing markers [51]. The concomitant reduced immune surveillance and phagocytosis levels as well as the lower vaccination efficacy render individuals prone to infections, endogenous retrovirus expression [52], cancer, autoimmune diseases and neurodegenerative or other age-dependent immune-mediated diseases [47,53,54,55]. Immunosenescence is a multifactorial phenomenon that is under the impact of genetic (e.g., high ROS production, telomere shrinkage, altered DNA repair and transcription factor induction [50]), immunological and environmental factors, such as stress, medication, nutrition, exercise, previous exposure to pathogens and co-morbidities [54,56]. 

Immunosenescence along with chronic systemic inflammation lead to inflammageing, a persistent type of systemic inflammation linked to higher mortality in seniors [21]. Inflammageing pillars include metabolic dysregulation, increased activation of macrophages that produce pro-inflammatory cytokines (such as IL-1β, IL-6 and TNF [49,54]), gut microbiota metabolites and the loss of neutrophils’ ability to produce enough peroxide and nitric oxide [49]. Inflammageing has been strongly correlated with age-dependent chronic diseases, from osteoporosis to metabolic and neurodegenerative diseases, possibly due to changes in glial cells that lead to neuroinflammation [47,49,57]. Specifically in MS patients, the exacerbated immune responses may accelerate CNS ageing, accompanied by escalating neurodegenerative events such as synaptopathy and synaptic plasticity dysregulation [57]. The resolution of inflammation is necessary to avoid such detrimental pathological neurodegenerative conditions and it normally involves both innate and adaptive immune responses [58]. In this regard, a potential therapeutic target may be the choroid plexus, which appears to be dysregulated in various neurodegenerative disorders [58] as well as the gut microbiota. Since dysbiosis is involved in immunological, metabolic and brain functions affecting one’s health condition during ageing, healthy diets rich in vegetables, whole grains, nuts and fish could help with achieving a reduction in inflammatory mediators [40]. Overall, immunosenescence and inflammageing represent the interplay between all branches of the immune system that affect each other and are an indispensable part of the whole ageing process and a pillar of geroscience, a new inspiring field of study [47,56].

## 3. Ageing, CNS Architecture and Function

As time passes, brain regions responsible for higher cognitive functions exhibit lower coordination, probably due to a gradual loss of neurons and myelinated fibers that mediate the connectivity of different regions or alterations in synaptic structure and function [59]. Studies on the brains of rats, mice, monkeys and humans have revealed variations in synaptic gene expression during ageing [35]. Moreover, an underexpression of genes of inhibitory neurotransmission through GABA (γ-aminobutyric acid) in the prefrontal cortex of monkeys and humans affects inhibitory and excitatory neurotransmission, therefore increasing the possibilities for neurodegeneration [35]. As the nervous system interacts with every organ, more and more evidence supports its role as a general ageing modulator [59].

Seniors often exhibit changes in their nervous system that are capable of modifying their brain structure and manifest as deficits of their cognitive functions [60]. During ageing, abundant alterations take place in the brain, including widely (global atrophy) or locally (focal atrophy) distributed cerebral or brain atrophy of the grey and white matter, that affect cognitive functions [61] and are estimated to occur at a rate of 0.5% of brain volume annually in individuals over sixty years old [62]. The decrease in dendrites and synapses, the elimination of myelinated nerve fibers and neuronal death are possible factors entangled in brain atrophy causing a different burden in each individual patient [63]. Brain atrophy seems to escalate gradually in specific brain areas such as the hippocampus, precentral gyrus, entorhinal cortex and putamen [64,65], while it is less intensified in the caudate, thalamus, amygdala and nucleus accumbens [61]. 

White matter lesions (WMLs) also seem to increase further with age, varying from around 5% up to 90% [66,67]. WMLs may be detected periventricularly or deeper in the frontal lobes, and subsequently in parietal and occipital lobes affecting executive, memory or motor functions depending on their location [61,68]. Furthermore, older people appear to have expanded perivascular spaces (PVSs) [69] mainly in the basal ganglia, the centrum semiovale and the midbrain, possibly due to local tissue atrophy, reduced interstitial fluid drainage and gliosis [61]. PVSs are often linked to cognitive decline, cerebral small vessel disease, lupus [69] and MS [70]. Additionally, the elderly slowly become more susceptible to microvascular changes such as microinfarcts and microbleeds [71] that may lead to functional and cognitive deficits [61].

Collective studies regarding the aforementioned lesions have elucidated their synergistic effect in cognitive impairment. Such associations were detected quantifying WMLs and atrophy, WMLs and PVS [61,72], WMLs and microbleeds, the levels of PVS, atrophy, subcortical infarcts and microbleeds [61,73]. Generally, a phenomenally benign change, such as PVS, that initially would not seem to have a major impact because of compensatory mechanisms, with time progression (when co-existing with more cumulative damage, such as WMLs) leads to the disruption of the networks and proceeds to the atrophy of the nervous system, which eventually manifests as cognitive impairment or dementia [61].

Ageing induces various alterations in the underlying basic structural components of the nervous system, including dendrites, axons and their connections via synapses [60]. As organisms age, a loss of axons and synapses increases in a time-dependent manner and contributes to earlier symptom manifestation of neurodegenerative disorders. A study conducted on individuals of 20–80 years of age revealed an up to 45% decrease in myelinated axons [74] possibly due to microischemic lesions caused by small cerebral blood vessels, which lead to oligodendrocyte and myelin disruption and susceptibility to cognitive decline [75]. In rhesus monkeys, myelinated axons were 20% lower in older monkeys, who also had lower cognitive performance in behavioral tests [76]. Notably, unmyelinated axons seem to be more prone to destruction as time progresses [74]. The main cause of age-related nerve fiber reduction seems to be myelin disruption rather than a major oligodendrocyte loss [75]. 

Elderly animals are also vulnerable to optic nerve damage as retinal ganglion cell (RGC) axons decrease [75]. Another interesting observation regarding axonal loss is that older people and rodents present reduced parahippocampal white matter, with the perforant pathway being implicated as well [75]. Considering that axons are pivotal for neuronal survival, some axonal arbors seem to be abrogated with the remaining arbors being able to maintain nerve cell homeostasis, hence reducing cell death rates [74]. In agreement with this, there is proof of lower cell death levels and arbor thinning in aged mice [77].

Furthermore, a loss in synaptic connections and the dendritic spine might occur due to presynaptic or postsynaptic events. Autophagosomes in presynaptic terminals are important in debris removal and are also involved in age-dependent neurodegenerative diseases such as Parkinson’s disease (PD) and Huntington’s disease (HD) [78]. The prefrontal cortex and hippocampus present different ageing patterns [79]. In line with this, studies on monkeys revealed a greater loss of thin spines in the prefrontal cortex, rather than in the hippocampus which tends to lose complex synapses that are important for the proper regulation of established memories and learning [60,74,79]. Another study reported a loss of hippocampal synapses in aged rats, resulting in spatial learning deficits and general cognitive impairment [80]. Remarkably, despite the loss in synapses that follows axonal loss, reinnervation might occur between some neurons and skeletal muscle, as reported in studies on mice [74,81]. Therefore, alterations in synapses that affect the structure and the function of neurons could possibly lead to neuronal senescence and destructive consequences for neuronal networks [74].

## 4. Ageing and CNS Neurodegenerative Disorders

Every single organism possesses an independent rate in gene expression alteration in the course of ageing, a concept that dictates the onset of age-related brain abnormalities and neurodegenerative diseases, such as AD and PD [8]. Advances in the research field have proposed various biomarkers associated with ageing and its related diseases, including imaging biomarkers and fatty acid metabolism for brain disorders such as AD, serum or urine metabolomics for cognitive decline, transcriptomics for mental illness and epigenetic clocks. The latter, based on DNA methylation, are able to determine a sample’s age which can be increased in those with PD, HD or AD disease, stroke, amyotrophic lateral sclerosis and MS [82,83]. Neurogranin, a small protein involved in synaptic plasticity and regeneration, has also been suggested as a possible biomarker for neuro-HIV, depression, schizophrenia, AD, PD and Creutzfeldt–Jakob disease [84], and telomere length was also formerly used as an ageing biomarker [82].

Although the etiology of the most common neurodegenerative disease, AD, has not yet been elucidated, it has been corelated with ageing, a risk factor of the disease that presents neuroinflammation and amyloid protein accumulation [85,86]. Ageing and AD appear to have various similarities in their phenotypic manifestations and underlying mechanisms. In both cases, declines in memory and other cognitive functions are correlated with neuronal networks found in the mediotemporal lobe with hippocampal formation and the parahippocampal region being especially affected [82]. Furthermore, deficits in telomere maintenance both in ageing and AD patients have been associated with the appearance of the disease’s characteristic pathological findings, inflammation and cognitive deficits [87]. Similarly, the epigenetic changes that occur over time affect memory, learning and behavior, and increase the risk of developing the specific disease [88]. Additionally, stem cell exhaustion has also been linked to AD and in the case of CNS stem cells it has been associated with impaired cognition reflecting limited neurogenesis in the dentate gyrus of patients. In addition, the similarity of immune responses in AD and the ageing CNS is consistent with the view that the changes occurring in the disease are affected by the manifestation of immunosenescence mechanisms [85]. Additionally, mitochondrial dysfunction, excessive oxidative damage and the deregulation of SIRT signaling are other common traits that have been demonstrated. Evidence from rodent models and neuronal cell cultures supports the ability of SIRT1 to ameliorate AD-like disorders probably through inducing disintegrin protein expression via the ADAM10 gene and through preserving metabolic homeostasis and nervous cells’ survival [86]. 

Diminished dendrites, synapses and even the axons of cortical neurons are some of the first traits of the aged human brain that are also seen in those with AD and result in the dysregulation of the functions that are under the control of cortical systems. As senescence has an impact on cellular maintenance through the downregulation of intermediary metabolism, which could lead to excessive oxidative damage and the disturbance of calcium signaling in the brain, it negatively affects the numbers of synapses and cortical functions [89]. The latter could be intensified by lifestyle choices that are associated with metabolism, such as unhealthy diet and lack of exercise, which also contribute to earlier signs of brain ageing and AD. Additionally, it was supported that neuroplasticity of the ageing brain could further contribute to AD pathology as abnormal neuronal sprouting responses were also involved in the appearance of AD neuronal plaques [89]. Furthermore, the resemblance of immune responses in AD and the aged CNS supports the notion that alterations in those with AD could be happening due to immunosenescence [85]. The ageing brain is affected amongst others by microglial senescence leading to further neurodegeneration and sporadic AD cases. Notably, modified microglia structure and decreased arborization have been observed in the ageing brain and in AD patients [85,90]. A recent study suggests that an AD-associated astrocytic population can appear during the first stages of the disease and increase further with age. Such astrocytes have been detected both in ageing human brains and aged wild-type mice, underlying their association with age-related and genetic factors [91].

Ageing has also been accused as being the main risk factor for PD, affecting the survival of dopaminergic neurons and the concomitant manifestations of the disease [92]. Age advancing is linked to more severe cognitive impairment and severe postural, gait and motor impairments in patients with PD and, notably, a later disease onset is associated with a more serious clinical course [93]. Neuroanatomical modifications, the senescence of cells, alterations of glial cells, mitochondrial dysfunction, excessive α-synuclein levels, inflammation along with oxidative stress and gut microbiome deregulation are only some of the common traits found in ageing and those with PD [94]. The main findings of PD are a loss of dopaminergic neurons in the substantia nigra pars compacta and intracellular accumulation of α-synuclein aggregates, known as Lewy bodies, changes that appear during ageing as well [94,95]. Dopaminergic neuron susceptibility in PD could occur due to extensive dependance on Ca^2+^ channels to sustain their activity with age, which could increase metabolic stress on mitochondria, cellular ageing and death [96]. Changes occurring in the neuronal microenvironment, such as in the phenotype of oligodendrocytes, astrocytes, microglia and vascular cells, also affect neuronal fate during ageing [8]. Some of these factors are capable of causing neuronal loss individually, but during ageing it is their combined action that renders substantia nigra neurons vulnerable and leads to their loss [92]. With increasing age, telomere shortening, epigenetic changes and the deregulation of DNA repair mechanisms further affect the function of the dopaminergic axis, multiplying the chances of PD appearance [97]. 

Changes regarding stem cell proliferation, differentiation, survival and neurite outgrowth and spine formation were also linked to PD. Decreased neurogenesis in adult stem cell niches, such as the hippocampus and subventricular zone (SVZ), could be related to depression and memory processing symptoms of PD [98]. An essential pathway for dopaminergic neurogenesis during development and ageing and for SVZ plasticity is the wingless-type mouse mammary tumor virus integration site (Wnt)/β-catenin (WβC) pathway, which also happens to be a risk factor of great importance for PD [99]. Restoring the WβC-signaling pathway and, thus, reactivating altered neurogenic niches that are affected by factors such as ageing, oxidative stress and inflammation could serve as a therapeutic approach to treat the effects of ageing on the PD brain [99]. Notably, inflammageing sustained by various cells such as neurons, microglia, leukocytes and astrocytes that can enter the CNS was also suggested to be linked with PD [94] (Figure 1).

## 5. Early and Late CNS Ageing Processes: The Case of MS

A triggering factor of relapsing remitting MS (RRMS) progression seems to be ageing, as patients with a later RRMS onset exhibit increased inflammation and oxidative stress in the CNS, modified synaptic plasticity and generally mild disease activity [100]. Additionally, telomere shrinkage has been linked to disability, independent of chronological age, leading to the hypothesis that biological ageing may be a factor that affects neurological injury in the case of MS [101]. An association has also been established between several factors, such as reduced physical activity, adolescent obesity and vitamin D deficiency, with telomere shrinkage, as well as a higher risk of MS. This indication is in agreement with the fact that the mechanisms underlying the disease are linked to cellular ageing and senescence which, in general, are further fueled by exacerbated inflammation and oxidative stress [102]. Over time, inflammation and oxidative stress lead to senescent cell accumulation, subsequently resulting in neurodegeneration and secondary progression of MS [103,104]. Ageing might also be involved in progression independent of relapse activity (PIRA) [100]. Generally, the activated neuroinflammation mechanisms might lead to synaptopathy, synaptic plasticity deterioration and, therefore, motor and cognitive deficits that manifest in different stages of life depending on the affected brain areas [57,105]. 

Synaptopathy in the CNS appears as a characteristic in both MS and its animal model EAE, concurrent with neurodegeneration [106]. In MS and EAE, various pro-inflammatory cytokines, such as tumor necrosis factor alpha (TNF-a), are secreted by infiltrating lymphocytes, activated microglia and astroglia promoting neurodegeneration, while acute inflammation is still active [57,107]. Such cytokines affect synaptic homeostasis via alterations in the glutamate receptors and ion channels [57]. In the course of time, brain milieu become progressively susceptible to pro-inflammatory cytokines which affect hippocampal neurogenesis and synaptic plasticity and are correlated with long-term memory induction and cognitive impairment in the elderly, along with miscellaneous neurobehavioral complications due to hyperactive microglia and neuroinflammation [108].

Different phases or phenotypes of MS have been linked to changes in brain plasticity affecting the overall clinical course. Brain plasticity collectively refers to the mechanisms that brain neural circuits rely on to counterbalance dysfunctional or unessential synapses [57] and properly respond to environmental stimuli by altering its functions or structure [74,109]. Neuroplasticity offers the capability to alter interneuron connections and generate new circuits, depending on the nervous system requirements. Plasticity is essential for learning and brain recovery in previously lesioned areas through structural remodeling, e.g., via axonal sprouting [109]. 

Ageing seems to be a factor that reduces plasticity as indicated in experimental animal models of CNS injury, such as in spinal cord hemisection. In such instances, remodeling occurs less frequently, and in the peripheral nerves of rats, regeneration might be reduced due to low myelin debris elimination speed or the compromised transport of materials for axon recovery. As age progresses, thin spines of the prefrontal cortex appear to be reduced and so does the ability to modify the synapses affecting learning, memory and long-term potentiation (LTP) [74], which is one of the mechanisms of brain plasticity that restores the excitability of synaptic injured neurons. The stimulation of the nervous system has been studied to determine how brain activity, emotional, social and other behaviors are modified with age [110]. In addition to this, neurogenesis has been proposed as another mechanism of neuroplasticity that is enhanced by activities such as learning. 

As far as we are aware, neurogenesis in the adult brain occurs in distinct stem cell niches in the ventricular and subventricular zones and in the subgranular zone of the hippocampus, regarding rodents, primates and humans [111,112]. Recent experimental data have suggested areas beyond the aforementioned zones where NSCs can be observed, such as the cerebral cortex, amygdala, substantia nigra, hypothalamus, striatum, septum and spinal cord [113]. Decreased neurogenesis in all of these regions may have an impact on synaptic plasticity, restoration potential and neurodegenerative disorders. SIRTs, which regulate synaptic plasticity and neurogenesis in adults, are also involved in the regulation of cognitive decline that accompanies ageing. SIRT1-deficient mice have been reported to present declines in hippocampal-dependent memory that are correlated with LTP [5,114]. In addition to this, SIRT1 can restrict neural stem cell (NSC) differentiation, controlling adult neurogenesis, and along with SIRT2 it can lead NSCs towards an oligodendrocytic fate. Generally, reduced hippocampal neurogenesis is associated with deficits in learning, memory and in LTP in the dentate gyrus of the hippocampus [5]. 

With regard to MS, the reparatory process of remyelination appears to occur at a lower rate in the elderly; therefore, demyelinating events and relapses in older patients can be more damaging than in young people [74]. Normally, the inhibition of oligodendrocyte differentiation is downregulated and histone deacetylases are recruited in order to promote remyelination; however, in aged brains these processes are inefficient, leading to the increased presence of transcriptional inhibitors and the prevention of myelin gene expression [44]. Moreover, in several cases, through the course of disease progression, the relapsing remitting pattern shifts to cumulative demyelination and permanent neurological deficits. In parallel, neuronal ageing also contributes to this shift and the remaining healthy neurons try to counterbalance the inflamed microenvironment via enlargement and axon arborization. Age-related changes, e.g., diminished vascularization, also contribute to the demise of healthy neuronal tissue [74]. 

Various changes in plasticity mechanisms that are associated with cognitive functions, such as a reduction in synapse numbers, occur in older animals. Data derived from electron microscopy observation of the perforant path granule cell synapse revealed a significant reduction in the axodendritic synapse numbers of the dentate gyrus in old rats, while decreased perforated synapses in the medial perforant path granule cell synapse have been linked to spatial memory impairment. With age progression, non-perforated and perforated axospinous synapses appear to be reduced and specifically a loss in functional hippocampal perforated synapses may be involved in cognitive decline during ageing and plasticity mechanisms. The estimation of LTP and long-term depression (LTD) reflects the impact of altered morphology and the function of synaptic connections on plasticity during ageing [65]. 

As brain ageing and neurodegeneration are highly associated with synaptic dysfunction, astrocytic perisynaptic processes, which are dynamic structures, also have an impact on structure and function that underlines their role in plasticity, depending on stimuli such as diet. Experimental data obtained from studying the morphology of astrocytes in sections of the mouse hippocampus from three different groups (young, adult and old) correlate astrocytic physiology with age resulting in a reduction in astrocytic coupling, size and perisynaptic processes’ volume [115]. Overall, the hippocampus appears to be significantly altered during ageing. The alterations that appear in glycolysis as time progresses may lead to AD as synaptic plasticity decreases. Therefore, it has been proposed that metabolic changes in neurons could be a factor linked to the dysregulated expression of the proteins that participate in brain plasticity mechanisms [116].

Age-related changes are additionally observed in innate and adaptive immune responses [117]. With regard to innate immunity, natural killer cells are increased, although with limited efficacy [50,118], dendritic cell functions are decreased [50,119] and neurophils and macrophages present a reduction in chemotaxis, pathogen killing and phagocytosis, respectively [50]. T and B lymphocytes of the adaptive immunity present a reduction in naïve lymphocytes and an increase in memory subsets, as well as changes in signaling, especially of the T cell [119]. In the case of MS, these alterations may have an additive effect on the impact of immunomodulatory treatments by increasing the risk of infections, cancer and other age-related adverse events.

## 6. Ageing, Comorbidities and DMTs

Age is a significant factor for MS-related disease progression, independently of initial disease type, age of onset, duration of the disease and gender [120]. Moreover, older age at MS onset is an independent factor of poor prognosis [121]. The age distribution in those with MS is shifting to older age groups, with a peak prevalence for the middle-aged in several ethnic populations [122,123]. As the effect of age in those with MS may evidently affect clinical outcomes, most clinical trials regarding the efficacy and safety of disease modifying treatments (DMTs) are performed in the young MS population; people with MS (PwMS) over 55 years old are rarely included [124]. In addition, the prevalence of comorbid conditions is increasing with age and, due to these conditions, many people with MS are excluded in clinical studies [125]. Therefore, the data regarding the efficacy and safety of DMTs in aged PwMS are scarce.

The presence of cardiovascular comorbidities, in particular, is often observed in elderly PwMS and is strongly correlated with faster rates of neurodegenerative processes, as reflected by more of a rapid accumulation of disabilities [126], lower cognitive performance and reduced total and regional brain volumes [127]. Therefore, in most studies, the presence of such a comorbidity is an exclusion criterion, since it may interfere with the drug’s safety and/or efficacy results assessed in a clinical trial. As a consequence, there is limited knowledge regarding the impact of cardiovascular comorbidities, commonly seen in this age group, with respect to the disease course and the potential drug interactions between DMTs and medications administered for cardiovascular comorbidity.

Moreover, as the use of MRI becomes increasingly available in most countries and physicians’ awareness of MS is improved, an increased incidence of MS diagnosis is evident [128,129]. The first patients who were diagnosed with MS and were exposed to MS-specific DMTs are expected to be in their late 50s and 60s by now. This fact contributes to the limited knowledge on the impact of DMTs in aged populations, thus highlighting the need for studies that include these age groups. Indeed, many studies that focus on the degenerative processes of the disease are now including PwMS over 55 years old [130,131]. In addition, newer studies are now implementing age as a significant parameter that implicates the quantitative evaluation of MS-related disability [132]. Therefore, the proposed outcome measure termed “age related multiple sclerosis severity score” is increasingly used in order to determine the disability of people with MS. Additionally, this may enable the differentiation of physical limitations and symptoms attributed to neurodegenerative processes due to aging from those directly related to MS [133]. Furthermore, new tools and measures are addressing the need of accurately quantifying disability in MS [134].

Up until now, the limited experience in DMT use in elderly PwMS has mainly been recorded in real-world studies, while some data may have been extracted from the available randomized clinical trials (RCTs). Several meta-analyses have attempted to elucidate the impact of DMT use in this population. Wiedeman et al. [135] reviewed data from 38 RCTs and concluded that their efficacy is strongly correlated with age. According to their findings, most PwMS over 53 years old are not expected to benefit from any DMT. Schweitzer et al. reviewed data regarding adverse events presented in PwMS treated with high efficacy DMTs [136]. Their main finding was that the risk for adverse events such as infections/viral reactivations and cancer increases with age.

Given all of the above, the timing of DMT discontinuation in elderly PwMS is still a matter of research [137]. On the one hand, inflammatory activity, which is the main target of DMTs, is rarely seen in PwMS over 60 [48], while those who remain under DMT do not report any benefit [138]. On the other hand, clinicians may hesitate to discontinue DMTs due to the fear of rebound disease activity and the belief that disease remission is attributed to DMTs and not to the natural course of the disease [139]. Up until now, all available data have supported the discontinuation of DMTs in this population, provided that that no inflammatory activity is observed [48,135,136,138,139,140]; however, future studies should be specifically designed to address this question.

## Figures and Tables

**Figure 1 jcm-12-02255-f001:**
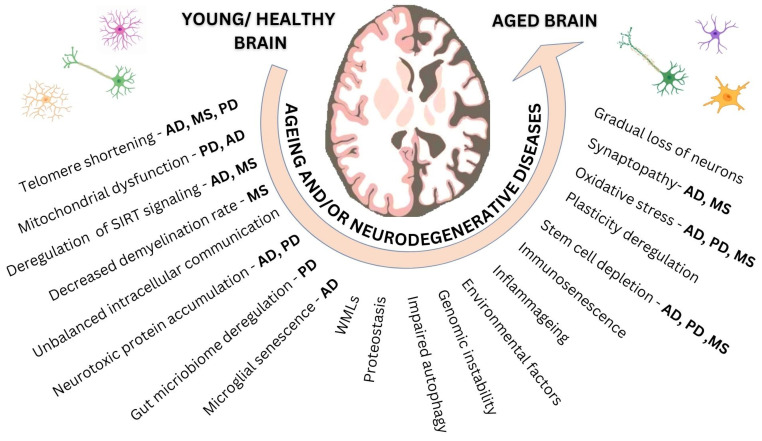
The hallmarks of ageing and the crosstalk between CNS and neurodegeneration. As time passes under the influence of molecular and cellular senescent mechanisms, brain cells are prone to inflammation and degeneration. Abbreviations: AD: Alzheimer’s disease, MS: multiple sclerosis, PD: Parkinson’s disease, SIRT: sirtuins, WMLs: white matter lesions.

## Data Availability

Not applicable.

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
