# Peer review of "CNS Ageing in Health and Neurodegenerative Disorders"

_jcm, 2023, doi:10.3390/jcm12062255_

Round 1

Reviewer 1 Report

The author reviewed the molecular changes and damage mechanism of central nervous system aging, and expounded the relationship between central nervous system aging and neurodegenerative diseases.

1. The author should add some latesed references, such as PMID: 36610399,PMID: 36336680.

2. The author should elaborate on the marker of central nervous system aging. As a terminal differentiated cell, can P16, p19 and p21 be used as markers of neuronal aging?

3.  The text of the article seems to be better organized. Cell metabolism, epigenetic regulation and some molecular pathways are inseparable in CNS aging. For example, cell glycolipid metabolism is regulated by m-TOR pathway. Another example is Sirtuin (SIRT), which is located on the mitochondria and closely related to metabolism. At the same time, it is also a deacetylase. Relevant content is not well connected and organized.

4.It is suggested that the author add 1-2 figures to more intuitively display the changes of CNS aging and the relationship with diseases.

Author Response

We wish to thank you for your important remarks, as they helped to improve our manuscript and we hope we have fulfilled your requests.

REVIEWER 1

1.The author should add some latest references, such as PMID: 36610399,PMID: 36336680.

We thank the reviewer for bringing these articles to our attention. The references were included appropriately (Section2.1 and Section 2.3)

2.The author should elaborate on the marker of central nervous system aging. As a terminal differentiated cell, can P16, p19 and p21 be used as markers of neuronal aging?

We would like to thank the reviewer for this remark. Despite the fact that there are available markers for the characterization of senescent cells, like P16, p19 and p21, they are linked mainly with neuronal development and not well studied regarding neuronal ageing. There are evidence for potential function of these CDK inhibitors to nervous system but further investigation is needed. In order to make a reference to these important markers a sentence was added in section 2.3.

3. The text of the article seems to be better organized. Cell metabolism, epigenetic regulation and some molecular pathways are inseparable in CNS aging. For example, cell glycolipid metabolism is regulated by m-TOR pathway. Another example is Sirtuin (SIRT), which is located on the mitochondria and closely related to metabolism. At the same time, it is also a deacetylase. Relevant content is not well connected and organized.

We would like to thank the reviewer for the fruitful comment. Indeed, there are several pathway-driven mechanisms which are universally utilized by a cell to regulate metabolism and function properly both in early or late-life homeostasis. For instance, sirtuins and mTOR are involved in the same longevity pathway. Importantly, resveratrol, an activator of sirtuins, antagonizes the mTOR/S6K pathway. However, disease conditions re-arrange most pathways and impede a clear distinction within an otherwise relevant content. Hence, we believe this presentation suits current needs with regards to aging and CNS neurodegenerative disorders.

4. It is suggested that the author add 1-2 figures to more intuitively display the changes of CNS aging and the relationship with diseases.

To address the reviewer’s comment, we added Figure 1 to the manuscript appropriately.

Reviewer 2 Report

In this work, the authors review the well-established link between ageing and neural dysfunction, with a particularly emphasis on MS, which has, to this point, not been as greatly explored as other age-related neuropathologies. However, the manuscript could possibly gain if this relationship, between ageing and MS, were explored to more detail.

The work starts with a general introduction to ageing (section 1), and then proceeds to a section focused on molecular and cellular events in CNS (section 2). Following, section 3 is dedicated to CNS architecture during ageing.

In section 2.1, genetic instability is discussed, but relevance for CNS ageing and neuropathologies is not established, while proteostasis is associated with AD, only.

The following section, 2.2, is dedicated to responses to damage explores secondary ageing hallmarks, but, similarly, refers to systemic ageing with no specific focus on the nervous system.

In section 2.3, the authors fully explore the tertiary hallmarks with an interesting focus on brain health and homeostasis, even covering cell types other than neurons; this depth is perhaps missing in sections 2.1 and 2.2.

Section 4 is a particularly interesting section of the manuscript, focused on ageing and neurodegenerative pathologies. The authors particularly focus on AD and PD, and in section 5 they follow through to a refreshing take, in which MS in ageing, and disease modifying treatments (section 6) are discussed. These sections are the most interesting ones in the review, and they should be expanded. Perhaps sections 1-3 could be combined in a single, extensive, introductory section, discussing the hallmarks of ageing in the brain, before proceeding to associated pathologies: the well-studied and established ones (i.e. AD, PD, etc.), and the ones somewhat overlooked.

These focused on MS, should be expanded with bigger emphasis on recent literature; no papers are cited in the last two years, with the average cited literature dating to before 2016. For example, and relevant to early sections of the manuscript, ageing-associated genomic instability/telomere shortening has recently, in recent years, been linked to MS (Krysko et al., 2019; Habib et al., 2020; Hecker et al., 2021).

Finally, the work would greatly be improved with the addition of a scheme or a figure, illustrating the message it is trying to convey.

Minor revisions:

The existence of a subsection 3.1 is redundant, since it is the only one in section 3. The paragraph before it could perhaps work as an introduction to a continuous section 3.

Spell check is required, in terms of punctuation; some missing comas, particularly, needlessly difficult the reading.

Author Response

We wish to thank you for your important remarks, as they helped to improve our manuscript and we hope we have fulfilled your requests.

REVIEWER 2

1. In section 2.1, genetic instability is discussed, but relevance for CNS ageing and neuropathologies is not established, while proteostasis is associated with AD, only. The following section, 2.2, is dedicated to responses to damage explores secondary ageing hallmarks, but, similarly, refers to systemic ageing with no specific focus on the nervous system. In section 2.3, the authors fully explore the tertiary hallmarks with an interesting focus on brain health and homeostasis, even covering cell types other than neurons; this depth is perhaps missing in sections 2.1 and 2.2

We thank the reviewer for this observation. Section 2 serves as an introductory section with references to the nervous system in order to point out the universal hallmarks of ageing. A change of Section 2 title seems to serve our initial intention, as the following sections focus on nervous system ageing. The title changed from “Ageing at the molecular and the cellular level in the central nervous system (CNS)” to “Ageing at the molecular and the cellular level” and we included additional text with the appropriate references.

2. Section 4 is a particularly interesting section of the manuscript, focused on ageing and neurodegenerative pathologies. The authors particularly focus on AD and PD, and in section 5 they follow through to a refreshing take, in which MS in ageing, and disease modifying treatments (section 6) are discussed. These sections are the most interesting ones in the review, and they should be expanded. Perhaps sections 1-3 could be combined in a single, extensive, introductory section, discussing the hallmarks of ageing in the brain, before proceeding to associated pathologies: the well-studied and established ones (i.e. AD, PD, etc.), and the ones somewhat overlooked. These focused on MS, should be expanded with bigger emphasis on recent literature; no papers are cited in the last two years, with the average cited literature dating to before 2016. For example, and relevant to early sections of the manuscript, ageing-associated genomic instability/telomere shortening has recently, in recent years, been linked to MS (Krysko et al., 2019; Habib et al., 2020; Hecker et al., 2021).

We appreciate the reviewer’s suggestion. Sections 4,5 and 6 were revised and expanded and the corresponding references were included as requested. Changes are highlighted. The number of the references has been increased from 125 to 143.

3. Finally, the work would greatly be improved with the addition of a scheme or a figure, illustrating the message it is trying to convey.

To address the reviewer’s comment, we added Figure 1 to the manuscript appropriately.

Minor revisions:

-The existence of a subsection 3.1 is redundant, since it is the only one in section 3. The paragraph before it could perhaps work as an introduction to a continuous section 3.

We thank the reviewer for this remark. Subsection 3.1 was included in the introduction part of section 3.

-Spell check is required, in terms of punctuation; some missing comas, particularly, needlessly difficult the reading.

We have thoroughly rechecked our manuscript and performed the necessary changes appropriately.

Round 2

Reviewer 2 Report

I thank the authors for their revisions, which I believe to have improved the manuscript.